# Modeling Adversarial Noise for Adversarial Defense

## Abstract

Deep neural networks have been demonstrated to be vulnerable to adversarial noise, promoting the development of defense against adversarial attacks. Motivated by the fact that adversarial noise contains well-generalizing features and that the relationship between adversarial data and natural data can help infer natural data and make reliable predictions, in this paper, we study to *model adversarial noise* by learning the transition relationship between adversarial labels (i.e. the flipped labels used to generate adversarial data) and natural labels (i.e. the ground truth labels of the natural data). Specifically, we introduce an instance-dependent *transition matrix* to relate adversarial labels and natural labels, which can be seamlessly embedded with the target model (enabling us to model stronger adaptive adversarial noise). Empirical evaluations demonstrate that our method could effectively improve adversarial accuracy.

## 1 Introduction

Deep neural networks have been demonstrated to be vulnerable to adversarial noise (Goodfellow et al., 2015; Szegedy et al., 2014; Jin et al., 2019; Liao et al., 2018; Ma et al., 2018; Wu et al., 2020a). The vulnerability of deep neural networks seriously threatens many decision-critical deep learning applications (LeCun et al., 1998; He et al., 2016; Zagoruyko & Komodakis, 2016; Simonyan & Zisserman, 2015; Kaiming et al., 2017; Ma et al., 2021).

To alleviate the negative affects caused by adversarial noise, a number of adversarial defense methods have been proposed. A major class of adversarial defense methods focus on exploiting adversarial examples to help train the target model (Madry et al., 2018; Ding et al., 2019; Zhang et al., 2019; Wang et al., 2019), which achieve the state-of-the-art performance. However, these methods do not explicitly model the adversarial noise. The relationship between the adversarial data and natural data has not been well studied yet.

Studying (or modeling) the relationship between adversarial data and natural data is considered to be beneficial. If we can model the relationship, we can infer natural data information by exploiting the adversarial data and the relationship. Previous researches have made some explorations on this idea. Some data pre-processing based methods (Jin et al., 2019; Liao et al., 2018; Naseer et al., 2020; Zhou et al., 2021), which try to recover the natural data by removing the adversarial noise, share the same philosophy. However, those methods suffer from the high dimensionality problem because both the adversarial data and natural data are high dimensional. The recovered data is likely to have *human-observable loss* (i.e., obvious inconsistency between processed examples and natural examples) (Xu et al., 2017) or contain residual adversarial noise (Liao et al., 2018).

To avoid the above problems, in this paper, we propose to model adversarial noise in the low-dimensional label space. Specifically, instead of directly modeling the relationship between the adversarial data and the natural data, we model the adversarial noise by learning the label transition from the *adversarial labels* (i.e., the flipped labels used to generate adversarial data) to the *natural labels* (i.e., the ground truth labels of the natural data). Note that the adversarial labels have not been well exploited in the community of adversarial learning, which guide the generation of adversarial noise and thus contain valuable information for modeling the well-generalizing features of the adversarial noise (Ilyas et al., 2019).

It is well-known that the adversarial noise depends on many factors, e.g., the data distribution, the adversarial attack strategy, and the target model. The proposed adversarial noise modeling method is capable to take into account of the factors because that the label transition is dependent of the adversarial instance (enabling us to model how the patterns of data distribution and adversarial noise affect label flipping) and that the transition can be seamlessly embedded with the target model (enabling us to model stronger adaptive adversarial noise). Specifically, we employ a deep neural network to learn the complex instance-dependent label transition.

By using the transition relationship, we propose a defense method based on Modeling Adversarial Noise (called MAN). Specifically, we embed a label *transition network* (i.e., a matrix-valued transition function parameterized by a deep neural network) into the target model, which denotes the transition relationship from mixture labels (mixture of adversarial labels and natural labels) to natural labels (i.e., the ground truth labels of natural instances and adversarial instances). The transition matrix explicitly models adversarial noise and help us to infer natural labels. Considering that adversarial data can be adaptively generated, we conduct joint adversarial training on the target model and the transition network to achieve an optimal adversarial accuracy. We empirically show that the proposed MAN based defense method could provide significant gains in the classification accuracy against adversarial attacks in comparison to state-of-the-art defense methods.

The rest of this paper is organized as follows. In Section 2, we introduce some preliminary information and briefly review related work on attacks and defenses. In Section 3, we discuss how to design an adversarial defense by modeling adversarial noise. Experimental results are provided in Section 4. Finally, we conclude this paper in Section 5.

## 2 PRELIMINARIES

In this section, we first introduce some preliminary about notation and the problem setting. We then review the most relevant literature on adversarial attacks and adversarial defenses.

**Notation.** We use *capital* letters such as $X$ and $Y$ to represent random variables, and *lower-case* letters such as $x$ and $y$ to represent realizations of random variables $X$ and $Y$, respectively. For norms, we denote by $\|x\|$ a generic norm. Specific examples of norms include $\|x\|_\infty$, the $L^\infty$ norm of $x$, and $\|x\|_2$, the $L^2$ norm of $x$. Let $\mathbb{B}(x, \epsilon)$ represent the neighborhood of $x$: $\{\tilde{x} : \|\tilde{x} - x\| \leq \epsilon\}$, where $\epsilon$ is the perturbation budget. We define the *classification function* as $f : \mathcal{X} \to \{1, 2, \ldots, C\}$. It can be parametrized, e.g., by deep neural networks.

**Problem setting.** This paper focuses on a classification task under the adversarial environment, where adversarial attacks are utilized to craft adversarial noise to mislead the predictions of the target classification model. Let $X$ and $Y$ be the variables for natural instances and natural labels (i.e., the ground truth labels of natural instances) respectively. We sample natural data $\{(x_i, y_i)\}_{i=1}^n$ according to the distribution of the variables $(X, Y)$, where $(X, Y) \in \mathcal{X} \times \{1, 2, \ldots, C\}$. Given a deep learning model based classifier $f$ and a pair of natural data $(x, y)$, the adversarial instance $\tilde{x}$ satisfies the following constraint:

$$f(\tilde{x}) \neq y \quad \text{s.t.} \quad \|x - \tilde{x}\| \leq \epsilon. \tag{1}$$

The generation of the adversarial instance is guided by the adversarial label $\tilde{y}$ which is different with the natural label $y$. Adversarial labels can be set by the attacker in target attacks, or be randomly initialized and then found by non-target attacks. Let $\widetilde{X}$ and $\widetilde{Y}$ be the variables for adversarial instances and adversarial labels respectively. We denote by $\{(\tilde{x}_i, \tilde{y}_i)\}_{i=1}^n$ the adversarial data drawn according to a distribution of the variables $(\widetilde{X}, \widetilde{Y})$. Our aim is to design an adversarial defense to correct the adversarial label $\tilde{y}$ into natural label $y$ by modeling the relationship between adversarial data (i.e., $\tilde{x}$ or $\tilde{y}$) and natural data (i.e., $x$ or $y$).

**Adversarial attacks.** Adversarial examples are inputs maliciously designed by adversarial attacks to mislead deep neural networks (Szegedy et al., 2014). They are generated by adding imperceptible but adversarial noise to natural examples. Adversarial noise can be crafted by multi-step attacks following the direction of adversarial gradients, such as PGD (Madry et al., 2018) and AA (Croce & Hein, 2020). These adversarial noise is bounded by a small norm-ball $\| \cdot \|_p \leq \epsilon$, so that their adversarial examples can be perceptually similar to natural examples. Optimization-based attacks such as CW (Carlini & Wagner, 2017b) and DDN (Rony et al., 2019) jointly minimize the perturbation $L_2$ and

a differentiable loss based on the logit output of a classifier. Some attacks such as FWA (Wu et al., 2020b) and STA (Xiao et al., 2018) focus on mimicking non-suspicious vandalism by exploiting the geometry and spatial information.

**Adversarial defenses.** The issue of adversarial examples promotes the development of adversarial defenses. A major class of adversarial defense methods is devoted to enhance the adversarial robustness in an adversarial training manner (Madry et al., 2018; Ding et al., 2019; Zhang et al., 2019; Wang et al., 2019). They augment training data with adversarial examples and use a min-max formulation to train the target model (Madry et al., 2018). However, these methods do not explicitly model the adversarial noise. The relationship between the adversarial data and natural data has not been well studied yet.

In addition, some data pre-processing based methods try to remove adversarial noise by learning a denoising function or a feature-squeezing function. For example, denoising based defenses (Liao et al., 2018; Naseer et al., 2020) transfer adversarial examples into clean examples, and feature squeezing based defenses (Guo et al., 2018) aim to reduce redundant but adversarial information. However, these methods suffer from the high dimensionality problem because both the adversarial data and natural data are high dimensional. For examples, the recovered data is likely to have human-observable loss (Xu et al., 2017) (i.e., the obvious inconsistency between processed examples and natural examples), and the recovered data may contain residual adversarial noise (Liao et al., 2018), which would be amplified by the target model and mislead the final prediction. To avoid the above problems, we propose to model adversarial noise in the low-dimensional label space.

## 3 Modeling adversarial noise based defense

In this section, we present the proposed *Modeling Adversarial Noise* (MAN) based defense method, to improve the adversarial accuracy against adversarial attacks. We first illustrate the motivation of the proposed defense method (Section 3.1). Next, we introduce how to model adversarial noise (Section 3.2). Finally, we present the training process of the proposed adversarial defense (Section 3.3).

### 3.1 Motivation

Studying the relationship between adversarial data and natural data is considered to be beneficial for adversarial defense. If we can model the relationship, we can infer clean data information by exploiting the adversarial data and the relationship. Processing the input data to remove the adversarial noise is a representative strategy to estimate the relationship. However, data pre-processing based methods may suffer from high dimensionality problems. To avoid the problem, we would like to model adversarial noise in the low-dimensional label space.

Adversarial noise can be modeled because it have imperceptible but well-generalizing features. Specifically, classifiers are usually trained to solely maximize accuracy for natural data. They tend to use any available signal including those that are well-generalizing, yet brittle. Adversarial noise can arise as a result of perturbing these features (Ilyas et al., 2019), and it controls the flip from natural labels to adversarial labels. Thus, adversarial noise contains well-generalizing features which can be modeled by *learning the label transition from the adversarial labels to natural labels*.

Note that the adversarial noise depends on many factors, such as data distribution, the adversarial attack strategy, and the target model. Modeling adversarial noise in label space is capable to take into account of these factors. Specifically, since that the label transition is dependent of the adversarial instance, we can model how the patterns of data distribution and adversarial noise affect label flipping, and exploit the modeling to correct the flipping of the adversarial label. In addition, we can model adversarial noise crafted by the stronger white-box adaptive attack, because the label transition can be seamlessly embedded with the target model. By jointly training the transition matrix with the target model, it can also be adaptive to adversarial attacks. We employ a deep neural network to learn the complex label transition from the well-generalizing and misleadingly predictive features of instance-dependent adversarial noise.

Motivated by the value of modeling adversarial noise for adversarial defense, we propose a defense method based on Modeling Adversarial Noise (MAN) by exploiting label transition relationship.

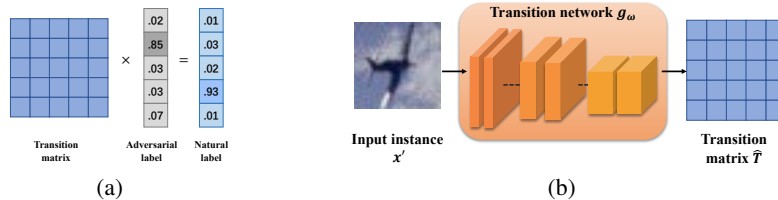

(a)                    (b)

Figure 1: (a). Infer the natural label by utilizing the transition matrix and the adversarial label. (b). The transition network $g_\omega$ parameterized by $\omega$ takes the instance $x'$ as input, and output the label transition matrix $\widehat{T}(x'; \omega) = g_\omega(x')$.

## 3.2 Modeling adversarial noise

We exploit a transition network to learn the label transition for modeling adversarial noise. The transition network can be regarded as a matrix-valued transition function parameterized by a deep neural network. The transition matrix, estimated by the label transition network, explicitly models adversarial noise and help us to infer natural labels from adversarial labels. The details are discussed below.

**Transition matrix.** To model the adversarial noise, we need to relate adversarial labels and natural labels in a explicit form (e.g., a matrix). Inspired by recent researches in label-noise learning (Xia et al., 2020; Liu & Tao, 2015; Xia et al., 2019; Yang et al., 2021; Wu et al., 2021; Xia et al., 2021), we design a label transition matrix, which can encode the probabilities that adversarial labels flip into natural labels. We then can infer natural labels by utilizing the transition matrix and adversarial data (see Figure 1(a)). Note that the transition matrix used in our work is different from that used in label-noise learning. The detailed illustration can be found in Appendix A

Considering that the adversarial noise is instance-dependent, the transition matrix should be designed to depend on input instances, that is, we should find a transition matrix specific to each input instance. In addition, we not only need to focus on the accuracy of adversarial instances, but also the accuracy of natural instances. An adversarially robust target model is often trained with both natural data and adversarial data. We therefore exploit the mixture of natural and adversarial instances (called mixture instances) as input instances, and design the transition matrix to model the relationship between the mixture of natural and adversarial labels (called mixture labels) to the ground-truth labels of mixture instances (called natural labels).

Specifically, let $X'$ denote the variable for the mixture instances, $Y'$ denote the variable for the mixture labels, and $Y$ denote the variable for the natural labels. We combine the natural data and adversarial data as the mixture data, i.e., $\{(x'_i, y'_i)\}_{i=1}^{2n} = \{(x_i, y_i)\}_{i=1}^{n} \cup \{(\tilde{x}_i, \tilde{y}_i)\}_{i=1}^{n}$, where $\{(x'_i, y'_i)\}_{i=1}^{2n}$ is the data drawn according to a distribution of the random variables $(X', Y')$. Since we have the ground-truth labels of mixture instances (i.e., natural labels), we can extend the mixture data $\{(x'_i, y'_i)\}_{i=1}^{2n}$ into a triplet form $\{(x'_i, y'_i, y_i)\}_{i=1}^{2n}$. We utilize the transition matrix $T \in [0, 1]^{C \times C}$ to model the relationship between mixture labels $Y'$ and natural labels $Y$. The transition matrix $T$ is dependent of the mixture instances $X'$. It is defined as:

$$T_{i,j}(X' = x') = P\left(Y = j \mid Y' = i, X' = x'\right), \tag{2}$$

where $T_{i,j}$ denotes the $(i, j) - th$ element of the matrix $T(X' = x')$. It indicates the probability of the mixture label $i$ flipped to the natural label $j$ for the input $x'$.

The basic idea is that given the *mixture class posterior probability* (i.e., the class posterior probability of the mixture labels $Y'$) $P(\boldsymbol{Y'} \mid X' = x') = [P(Y' = 1 \mid X' = x'), \dots, P(Y' = C \mid X' = x')]^\top$, the *natural class posterior probability* (i.e., the class posterior probability of natural labels $Y$) $P(\boldsymbol{Y} \mid X' = x')$ could be inferred by exploiting $P(\boldsymbol{Y'} \mid X' = x')$ and $T(X' = x')$:

$$P(\boldsymbol{Y} \mid X' = x') = T(X' = x')^\top P(\boldsymbol{Y'} \mid X' = x'). \tag{3}$$

We then can obtain the natural labels by choose the class labels that maximize the robust class posterior probabilities. Note that the mixture class posterior probability $P(\boldsymbol{Y'} \mid X' = x')$ can be estimated by exploiting the mixture data.

**Transition network.** we employ a deep neural network (called *transition network*) to estimate the label transition matrix by exploiting the mixture data $\{(x'_i, y'_i, y_i)\}_{i=1}^{2n}$. Specifically, as shown in

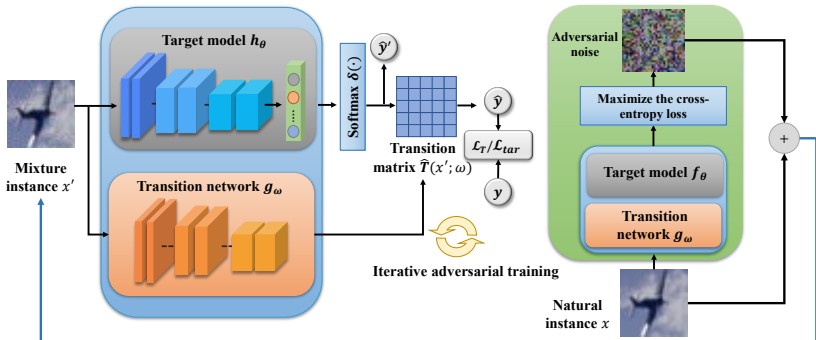

Figure 2: An overview of the training procedure for the proposed defense method. $\hat{\boldsymbol{y}}'$ and $\hat{\boldsymbol{y}}$ denote the probability of the estimated mixture label $\hat{y}'$ and the probability of the inferred natural label $\hat{y}$ respectively, i.e., $\hat{\boldsymbol{y}}' = \delta(h_\theta(x'))$ and $\hat{\boldsymbol{y}} = \hat{\boldsymbol{y}}' \cdot \widehat{T}(x'; \omega)$. $\boldsymbol{y}$ is $y$ in the form of a vector.

Figure 1(b), the transition network $g_\omega(\cdot)$ parameterized by $\omega$ takes the mixture instance $x'$ as input, and output the label transition matrix $\widehat{T}(x'; \omega) = g_\omega(x')$. We then can infer the natural class posterior probability $P(\boldsymbol{Y} \mid X' = x')$ according to Eq. 3, and thus infer the natural label. To optimize the parameter $\omega$ of the transition network, we minimize the difference between the inferred natural labels and the ground-truth natural labels. The loss function of the transition network is define as:

$$\mathcal{L}_T(\omega) = -\frac{1}{2n} \sum_{i=1}^{2n} \ell(\boldsymbol{y}_{\boldsymbol{i}}' \cdot \widehat{T}(x_i'; \omega), \boldsymbol{y}_{\boldsymbol{i}}), \tag{4}$$

where $\boldsymbol{y}_{\boldsymbol{i}}'$, $\boldsymbol{y}_{\boldsymbol{i}}$ are $y_i'$, $y_i$ in the form of vectors. $\ell(\cdot)$ is the cross-entropy loss between the inferred natural labels and the ground-truth natural labels, i.e., $\ell(\boldsymbol{y}_{\boldsymbol{i}}' \cdot \widehat{T}(x_i'; \omega), \boldsymbol{y}_{\boldsymbol{i}}) = \boldsymbol{y}_{\boldsymbol{i}} \cdot \log(\boldsymbol{y}_{\boldsymbol{i}}' \cdot \widehat{T}(x_i'; \omega))$.

### 3.3 TRAINING

Considering that adversarial data can be adaptively generated, and the adversarial labels are also depended on the target model, we conduct joint adversarial training on the target model and the transition network to achieve the optimal adversarial accuracy. We provide the overview of the training procedure in Figure 2.

For the transition network, we optimize its model parameter $\omega$ according to Eq. 4. For the target model, considering that the final inferred natural class posterior probability is influenced by the target model, we also use the cross-entropy loss between the inferred natural labels and the ground-truth natural labels to optimize the parameter $\theta$. The loss function for the target model $h_\theta$ is defined as:

$$\mathcal{L}_{tar}(\theta) = -\frac{1}{2n} \sum_{i=1}^{2n} [\boldsymbol{y}_{\boldsymbol{i}} \cdot \log(\hat{\boldsymbol{y}}_{\boldsymbol{i}}' \cdot \widehat{T}(x_i'; \omega))], \hat{\boldsymbol{y}}_{\boldsymbol{i}}' = \delta(h_\theta(x_i')), \tag{5}$$

where $\delta(\cdot)$ denote the softmax function. $\hat{\boldsymbol{y}}_{\boldsymbol{i}}'$ denote the mixture label in the form of a vector predicted by the target model.

To demonstrate the effectiveness of joint adversarial training, we conduct an ablation study by independently training the transition network with a fixed pre-trained target model (see Section 4.3). In addition, to prove that using mixture instances can achieve better adversarial accuracy compared to only using adversarial instances, we conduct an ablation study by training the proposed defense using only adversarial data (see Section 4.3).

The details of the overall procedure are presented in Algorithm 1. Specifically, for each mini-batch $\mathcal{B} = \{x_i\}_{i=1}^m$ sampled from natural training set, we first generate adversarial instances $\widetilde{\mathcal{B}} = \{\tilde{x}_i\}_{i=1}^m$ via a strong adversarial attack algorithm, and obtain the mixture mini-batch $\mathcal{B}' = \{x_i'\}_{i=1}^{2m}$ (i.e., mixture of $\mathcal{B}$ and $\widetilde{\mathcal{B}}$) with mixture labels $\{y_i'\}_{i=1}^{2m}$ (i.e., mixture of adversarial labels $\{\tilde{y}_i\}_{i=1}^m$ and natural labels $\{y_i\}_{i=1}^m$). Then, we input the mixture instances $\{x_i'\}_{i=1}^{2m}$ into the target model $h_\theta(\cdot)$ and output $\{\hat{\boldsymbol{y}}_{\boldsymbol{i}}'\}_{i=1}^{2m}$. We also input the mixture instances $\{x_i'\}_{i=1}^{2m}$ into the transition network $g_\omega(\cdot)$

---
**Algorithm 1** Training of Modeling Adversarial Noise (MAN) based defense

---
**Input:** Target model $h_\theta(\cdot)$ parametrized by $\theta$, transition network $g_\omega(\cdot)$ parametrized by $\omega$, batch size $m$, and the perturbation budget $\epsilon$;
1: **repeat**
2:     Read mini-batch $\mathcal{B} = \{x_i\}_{i=1}^m$ from training set;
3:     Craft adversarial instance $\{\tilde{x}_i\}_{i=1}^m$ at the given perturbation budget $\epsilon$ for each instance $x_i$ in $\mathcal{B}$;
4:     Obtain the mixture mini-batch $\mathcal{B}' = \{x_i'\}_{i=1}^{2m}$ with mixture labels $\{y_i'\}_{i=1}^{2m}$;
5:     **for** $i = 1$ to $2m$ (in parallel) **do**
6:         Forward-pass $x_i'$ through $h_\theta(\cdot)$ and obtain $\hat{\boldsymbol{y}}_{\boldsymbol{i}}'$;
7:         Forward-pass $x_i'$ through $g_\omega(\cdot)$ to estimate $\widehat{T}(x_i'; \omega)$;
8:         Infer the final prediction label by exploiting $\hat{y}_i'$ and $\widehat{T}(x_i'; \omega)$;
9:     **end for**
10:     Calculate $\mathcal{L}_T(\omega)$ and $\mathcal{L}_{tar}(\theta)$ using Eq. 4 and Eq. 5;
11:     Back-pass and update $\omega$ and $\theta$;
12: **until** training converged.

---

to estimate the label transition matrices $\{\widehat{T}(x_i'; \omega)\}_{i=1}^{2m}$. We next infer the final prediction labels by exploiting $\{\hat{\boldsymbol{y}}_{\boldsymbol{i}}'\}_{i=1}^{2m}$ and $\{\widehat{T}(x_i'; \omega)\}_{i=1}^{2m}$. Finally, we compute the loss functions $\mathcal{L}_T(\omega)$ and $\mathcal{L}_{tar}(\theta)$, and update the parameters $\omega$ and $\theta$. By iteratively conducting the procedures of adversarial example generation and defense training, $\omega$ and $\theta$ are expected to be adversarially optimized.

# 4    EXPERIMENTS

In this section, we first introduce the experiment setup including datasets, attack settings and defense settings in Section 4.1. Then, we evaluate the effectiveness of our defense against representative and commonly used $L^\infty$ norm and $L^2$ norm adversarial attacks in Section 4.2. Finally, we conduct ablation studies in Section 4.3.

## 4.1    EXPERIMENT SETUP

**Datasets.** We verify the effective of our defense method on two popular benchmark datasets, i.e., *CIFAR-10* (Krizhevsky et al., 2009) and *Tiny-ImageNet* (Wu et al., 2017). *CIFAR-10* has 10 classes of images including 50,000 training images and 10,000 test images. *Tiny-ImageNet* has 200 classes of images including 100,000 training images, 10,000 validation images and 10,000 test images. Images in the two datasets are all regarded as natural instances. All images are normalized into [0,1], and are performed simple data augmentations in the training process, including random crop and random horizontal flip. For the target model, we use ResNet-18 (He et al., 2016) for *CIFAR-10* and *Tiny-ImageNet*.

**Attack settings.** Adversarial examples for evaluating defense models are crafted by applying state-of-the-art attacks. These attacks include $L^\infty$ norm PGD (Madry et al., 2018), $L^\infty$ norm AA (Croce & Hein, 2020), $L^\infty$ norm TI-DIM (Dong et al., 2019; Xie et al., 2019), $L^\infty$ norm FWA (Wu et al., 2020b), $L^2$ norm CW (Carlini & Wagner, 2017b) and $L^2$ norm DDN (Rony et al., 2019). Among them, the AA attack algorithm integrates three non-target attacks and a target attack. Other attack algorithms belong to non-target attacks. The iteration number of PGD is set to 40 with step size 0.007. The iteration number of FWA is set to 20 with step size 0.007. The iteration numbers of $CW_2$ and DDN are set to 200 and 40 respectively with step size 0.01. For *CIFAR=10* and *Tiny-ImageNet*, the perturbation budgets for $L^2$ norm attacks and $L^\infty$ norm attacks are $\epsilon = 0.5$ and $8/255$ respectively.

**Defense settings.** We use three representative defense methods as the baselines: standard adversarial training AT (Madry et al., 2018), TRADES (Zhang et al., 2019) and MART (Wang et al., 2019). We use the ResNet-18 as the transition network for *CIFAR-10* and *Tiny-ImageNet*. The perturbation budget $\epsilon$ is set to $8/255$ for *CIFAR-10* and *Tiny-ImageNet*. The epoch number is set to 150 for our defense, and 100 for baselines. For fair comparisons, all the methods are trained using SGD with momentum 0.9 and an initial learning rate of 0.1. The weight decay is $2 \times 10^{-4}$ for *CIFAR-10*, and is $5 \times 10^{-4}$ for *Tiny-ImageNet*. For TRADES and MART, we set $\lambda = 6$. We use the $L^\infty$ norm non-target PGD-10 (i.e., PGD with iteration number of 10) with random start and step size $\epsilon/4$ to craft adversarial training data for all defenses

### 4.2 DEFENSE EFFECTIVENESS

**Defending against adaptive attacks.** A powerful *adaptive attack* strategy has been proposed to break defense methods (Athalye et al., 2018; Carlini & Wagner, 2017a). In this case, the attacker can access the architecture and model parameters of both the target model and the defense model (i.e., the transition network). We study the following attack scenarios for evaluating our defense.

***Scenario (i).*** Considering that the models in baselines are completely leaked to the attacker, for fair comparison, we utilize white-box adversarial attacks against the combination of the target model and the transition matrix. Similar to attacks against baselines, the goal of the non-target attack in this scenario is to maximize the distances between final predictions of our defense and the ground-truth natural labels. The adversarial instance $\tilde{x}$ is crafted by solving the following optimization problem:

$$\max_{\tilde{x}} \mathcal{L}(\tilde{\boldsymbol{y}} \cdot \widehat{T}(\tilde{x}; \omega), \boldsymbol{y}), \text{ subject to: } \|x - \tilde{x}\| \leq \epsilon, \tag{6}$$

where $\tilde{\boldsymbol{y}} = \delta(h_\theta(\tilde{x}))$ and $\mathcal{L}(\cdot)$ denote the specific loss function used by each attack. Similarly, we can craft adversarial examples via the target attack. The details are presented in Appendix B.

We combine this attack strategy with six representative adversarial attacks introduced in section *attack settings* to evaluate defenses. For training our defense, We use adversarial examples crafted by adaptive PGD-10 attack as adversarial training data. The average natural accuracy (i.e., the results in the second column) and the average adversarial accuracy of defenses are shown in Table 1.

The results show that our defense achieves superior adversarial accuracy compared with AT. This presents that our defense is effective. In addition, our method achieves significant improvements against more destructive FWA, $CW_2$ and DDN attacks (e.g., the adversarial accuracy is increased from 0.00% to 46.36% against $CW_2$). Note that the training mechanism in our method can be regarded as the standard adversarial training on the combination of the target network and the transition matrix, our method thus is applicable to different adversarial training methods. To avoid the bias caused by different adversarial training methods, we apply our method to TRADES and MART. As shown in Table 1, the results show that our method improves the adversarial accuracy. We find that using TRADES and MART would affect the performance against $L_2$-norm based attacks. We will further solve this issue in future work. The results on *Tiny-ImageNet* are shown in Appendix B.1.

In addition, we evaluate the adversarial accuracy of defense methods against white-box adaptive attacks on *CIFAR-10* by using the VggNet-19 as the target model and the transition network. The results are also shown in Appendix B.1.

Table 1: Adversarial accuracy (percentage) of defense methods against white-box adaptive attacks on *CIFAR-10*. The target model is ResNet-18. We show the most successful defense with **bold**.

| Attack | None | PGD-40 | AA | FWA-40 | TI-DIM | $CW_2$ | DDN |
|---|---|---|---|---|---|---|---|
| AT | **83.39** | 42.38 | 39.01 | 15.44 | 57.12 | 0.00 | 0.09 |
| MAN | 82.71 | **44.84** | **39.31** | **29.70** | **58.88** | **46.36** | **10.73** |
| TRADES | **80.68** | 49.38 | 44.43 | 23.92 | 57.45 | 0.00 | 0.16 |
| MAN_TRADES | 76.28 | **49.94** | **44.49** | **30.51** | **57.53** | **7.69** | **1.18** |
| MART | **76.15** | 50.93 | 44.37 | 25.63 | 59.23 | 0.00 | 0.02 |
| MAN_MART | 75.29 | **50.97** | **44.45** | **31.24** | **59.28** | **6.08** | **1.06** |

***Scenario (ii).*** In this scenario, we design an adversarial attack to destroy the crucial transition matrix of our defense. If the transition matrix is destroyed, the defense would immediately become ineffective. Since the ground-truth transition matrix is not given, we use the target attack strategy to craft adversarial examples. We choose an anti-diagonal identity matrix (see Figure 4 in Appendix B.2) as an example of the target transition matrix in the target attack. The optimization goal is designed as:

$$\max_{\tilde{x}} -\mathcal{L}_{mse}(T(\tilde{x}; \omega), T^*), \text{ subject to: } \|x - \tilde{x}\| \leq \epsilon, \tag{7}$$

where $\mathcal{L}_{mse}$ denote the *mean square error* loss. We just showed two examples for non-targeted and targeted attacks designed to the transition network in the above two scenarios. Note that different attacks can be designed, which however is beyond the scope of our work.

We use $L^\infty$ norm PGD-40 with this target attack strategy to evaluate the MAN based defense trained in scenario (i). The adversarial accuracy is 70.26% on *CIFAR-10*, which represents that our defense is effective against this adaptive target attack. This may be because that the attack in scenario (i) craft adversarial data for reducing the accuracy by destroying the transition matrix as well. The transition network adversarially trained on such adversarial data thus have good robustness against the attack designed for the transition matrix.

***Scenario (iii).*** We explore another adaptive attack scenario. In this scenario, the attack not only disturbs the final prediction labels, but also disturbs the output of the target model. This attack is called *dual adaptive attack*. The optimization goal of the non-target attack can be designed as:

$$\max_{\tilde{x}} \left[ \mathcal{L}(\tilde{\boldsymbol{y}} \cdot T(\tilde{x}; \omega), \boldsymbol{y}) + \mathcal{L}_{ce}(\tilde{\boldsymbol{y}}, \boldsymbol{y}) \right], \text{ subject to: } \|x - \tilde{x}\| \leq \epsilon, \tag{8}$$

where $\tilde{\boldsymbol{y}} = \delta(h_\theta(\tilde{x}))$, $\mathcal{L}(\cdot)$ denote the specific loss function used by each adversarial attack and $\mathcal{L}_{ce}(\cdot)$ is the cross-entropy loss of the target model, i.e. $\mathcal{L}_{ce}(\tilde{\boldsymbol{y}}, \boldsymbol{y}) = -\frac{1}{n} \sum_{i=1}^{n} \boldsymbol{y_i} \cdot \log(\tilde{\boldsymbol{y}})$. Similarly, we can craft adversarial examples via the target attack. The details are presented in Appendix B.3.

We combine this dual adaptive attack strategy with above six attacks to evaluate the proposed defense trained in scenario (i). As shown in Table 2, the results in the second row are the adversarial accuracy of our MAN based defense in this scenario, the results in the third row (Model$_T$) show that accuracy of the target model for adversarial instances. The results demonstrate that our defense method can infer natural labels by exploiting the instance-dependent transition matrix and adversarial labels, it provides effective protection against multiple attacks in this scenario.

We note that the adversarial accuracy of MAN in Table 2 is higher than that in Table 1. This may be because, for some input instances, the dual adaptive attack mainly uses the gradient information of the attack loss only against the target model, to craft adversarial noise for breaking the target model. Our MAN based defense can model the adversarial noise (i.e., learn the transition relationship) for the adversarial label predicted by the target model. The final natural label can be inferred, and the final adversarial accuracy thus can be improved.

Table 2: Adversarial accuracy (percentage) of our MAN based defense against dual white-box adaptive attacks (the second row) and against general attacks (the third row) on *CIFAR-10*. The target model is ResNet-18.

| Attack | None | PGD-40 | AA | FWA-40 | TI-DIM | CW | DDN |
|---|---|---|---|---|---|---|---|
| MAN | 82.71 | 70.91 | 39.41 | 64.18 | 72.87 | 46.37 | 38.74 |
| Model$_T$ | 8.64 | 0.34 | 2.50 | 0.64 | 0.63 | 2.04 | 0.64 |
| MAN$^\star$ | 89.01 | 81.07 | 79.90 | 80.02 | 61.93 | 77.89 | 77.82 |
| Model$_T^*$ | 88.98 | 0.00 | 0.00 | 0.00 | 10.45 | 0.00 | 0.00 |

**Defending against general attacks.** We also evaluate the effectiveness of the proposed defense by using general attacks, i.e., the adversarial attacks against the target model. We utilize above six adversarial attacks to evaluate the proposed MAN based defense. We use the adversarial instances crafted by non-target $L^\infty$ norm PGD-10 attack against the target model as the adversarial training data, to train our defense model. The adversarial accuracy of our MAN based defense is shown in Table 2 (MAN$^\star$). The adversarial accuracy of the target model (Model$_T^*$) is also shown on the fifth row in Table 2. It can be seen that our defense method achieve a great defense effect against adversarial examples which are crafted against the target model.

**Defense transferability.** The work in Ilyas et al. (2019) shows that any two target models are likely to learn similar mixture pattern, so modeling the adversarial noise which manipulate such features would apply to both target models for improving adversarial accuracy. We apply the MAN based defense which is trained for the ResNet-18 (ResNet), to two naturally pre-trained target models VGG-19 (VGG) (Simonyan & Zisserman, 2015) and Wide-ResNet 28×10 (WRN) (Zagoruyko & Komodakis, 2016) for evaluating the transferability of our defense. The performance on *CIFAR-10* is reported in Table 3. It can be seen that our defense method has effective transferability for providing cross-model protections against adversarial attacks. In addition, we transfer our defense model to VGG-19 for defending against adaptive attacks. The details can be found in Appendix B.4

Table 3: Adversarial accuracy (percentage) of our defense method for different target models on *CIFAR-10* against general adversarial attacks.

| Attack | None | PGD-40 | AA | FWA-40 | TI-DIM | CW$_2$ | DDN |
|--------|------|--------|-----|--------|--------|--------|-----|
| ResNet | 89.01 | 81.07 | 79.90 | 80.02 | 61.93 | 77.89 | 77.82 |
| VGG | 89.17 | 72.05 | 71.86 | 71.25 | 61.01 | 77.03 | 77.80 |
| WRN | 91.19 | 70.08 | 69.66 | 69.40 | 57.58 | 77.05 | 77.70 |

## 4.3 ABLATION STUDY

To demonstrate that the joint adversarial training on the target model and the transition network could achieve better defense compare to only training the transition network, we conduct an ablation study on our MAN based defense. Specifically, we use a naturally pre-trained ResNet-18 target model, and train the transition network independently. The model parameters of the target model are fixed in the training procedure. The defense model is called as MAN-. We use the adversarial examples crafted by $L^\infty$ norm PGD-10 against the target model as adversarial training data. Compared with the above MAN$^*$, the results shown in Figure 3(a) demonstrated the joint adversarial training manner can provide positive gain.

In addition, to show the effect of using mixture instances on defense performance, we only utilize the adversarial instances to retrain the defense model (called MAN$'$). The adversarial training data are crafted by $L^\infty$ norm PGD-10 with adaptive attack strategy. We use white-box adaptive attacks (i.e., Eq. 6) to evaluate the defenses. The results as shown in Figure 3(b) represent that the using mixture data can achieve better performance in defending adaptive attacks.

Moreover, to show that the improvement of our method has little relationship with the fact that the transition network introduces more model parameters, we conduct an experiment by using two parallel ResNet-18 as the target model. The details can be found in Appendix C.

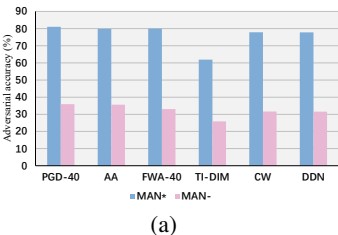
(a)

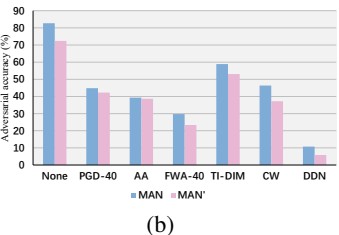
(b)

Figure 3: (a). Ablation study by independently training the transition network. The adversarial examples are crafted against the target model. (b). Ablation study by training the proposed defense using only adversarial data. The adversarial examples are crafted by white-box adaptive attacks.

## 5 CONCLUSION

Traditional adversarial defense methods typically focus on directly exploiting adversarial examples to remove adversarial noise or train an adversarially robust target model. In this paper, motivated by that the relationship between adversarial data and natural data can help infer clean data from adversarial data to obtain the final correct prediction, we study to model adversarial noise by learning the label transition relationship for using adversarial labels to improve adversarial accuracy. We propose a defense method based on Modeling Adversarial Noise (called MAN). Specifically, we embed a label transition network into the target model, which denote the transition relationship from mixture labels (i.e., mixture of adversarial labels and natural labels) to robust labels (i.e., natural labels). The transition matrix explicitly models adversarial noise and help us to infer robust labels. Considering that adversarial data can be adaptively generated, we conduct joint adversarial training on the target model and the transition network to achieve an optimal adversarial accuracy. The empirical results demonstrate that our defense method can provide effective protection against white-box general attacks and adaptive attacks. Our work provides a new adversarial defense strategy for the community of adversarial learning. In future, we will further optimize the MAN based defense method to improve its performance against white-box adaptive attacks.

**Ethics Statement** This paper doesn't raise any ethics concerns. This study doesn't involve any human subjects, practices to data set releases, potentially harmful insights, methodologies and applications, potential conflicts of interest and sponsorship, discrimination/bias/fairness concerns, privacy and security issues, legal compliance, and research integrity issues.

**Reproducibility Statement** To ensure the reproducibility of this work, we use the codes from the baseline method. We then conduct our experiments on this basis. Note that we report the *last* adversarial accuracy (i.e., the adversarial accuracy of the last epoch models obtained during training) in this work, instead of the *best* results.

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

# Appendices

## A    THE DIFFERENCE OF THE TRANSITION MATRIX BETWEEN OUR METHOD AND LABEL-NOISE LEARNING

In label-noise learning, the transition matrix is used to infer clean labels from given noisy labels. The transition matrix denotes the probabilities that **clean** labels flip into **noisy** labels (Xia et al., 2020; Liu & Tao, 2015; Yang et al., 2021). The label-noise learning methods utilize the transition matrix to correct the training loss on noisy data (i.e., the clean instance with the **noisy label**). Given the noisy class posterior probability, the clean class posterior probability can be obtained, i.e., $p(\boldsymbol{y}|x) = (T(x)^\top)^{-1}p(\boldsymbol{y}^*|x)$, where $\boldsymbol{y}^*$ denotes the noisy label in the form of a vector. Differently, in our method, the transition matrix is used to infer natural labels from observed adversarial labels. The transition matrix denotes the probabilities that **adversarial** labels flip into **natural** labels. We utilize the transition matrix to help compute the training loss on the adversarial instance with the **natural label**. For the observed adversarial class posterior probability, the natural class posterior probability can be obtained, i.e., $p(\boldsymbol{y}|\tilde{x}) = T(\tilde{x})^\top p(\tilde{\boldsymbol{y}}|\tilde{x})$. In addition, our method for the first time utilizes the transition matrix to explicitly model adversarial noise for improving adversarial robustness.

## B    DEFENDING AGAINST ADAPTIVE ATTACKS

### B.1    SCENARIO (I)

In scenario (i), the target attack solve the following optimization problem:

$$\max_{\tilde{x}} -\mathcal{L}(\tilde{\boldsymbol{y}} \cdot \widehat{T}(\tilde{x}; \omega), \boldsymbol{y}^*), \text{ subject to: } \|x - \tilde{x}\| \leq \epsilon, \tag{9}$$

where $\boldsymbol{y}^*$ is the target label in the form of a vector set by the attacker. The adversarial accuracy of defense methods against white-box adaptive attacks on *CIFAR-10* and *Tiny-ImageNet* are shown in Table 4 and Table 5 respectively.

Table 4: Adversarial accuracy (percentage) of defense methods against white-box adaptive attacks on *CIFAR-10*. The target model is ResNet-18. We show the most successful defense with **bold**. 'MAN_T' denotes the MAN_TRADES, and 'MAN_M' denotes the MAN_MART.

| Attack | None | PGD-40 | AA | FWA-40 | TI-DIM | CW$_2$ | DDN |
|---|---|---|---|---|---|---|---|
| AT | **83.39±0.95** | 42.38±0.56 | 39.01±0.51 | 15.44±0.32 | 57.12±0.71 | 0.00±0.00 | 0.09±0.03 |
| MAN | 82.71±0.62 | **44.84±0.42** | **39.31±1.13** | **29.70±0.41** | **58.88±0.49** | **46.36±1.32** | **10.73±0.67** |
| TRADES | **80.68±0.63** | 49.38±0.56 | 44.43±0.45 | 23.92±0.51 | 57.45±0.30 | 0.00±0.00 | 0.16±0.02 |
| MAN_T | 76.28±0.59 | **49.94±0.37** | **44.49±0.63** | **30.51±0.43** | **57.53±0.36** | **7.69±0.51** | **1.18±0.20** |
| MART | **76.15±0.91** | 50.93±0.72 | 44.37±0.83 | 25.63±0.88 | 59.23±0.39 | 0.00±0.00 | 0.02±0.01 |
| MAN_M | 75.29±0.76 | **50.97±0.63** | **44.45±0.79** | **31.24±0.57** | **59.28±0.37** | **6.08±1.69** | **1.06±0.23** |

Table 5: Adversarial accuracy (percentage) of defense methods against white-box adaptive attacks on *Tiny-ImageNet*. The target model is ResNet-18. We show the most successful defense with **bold**. 'MAN_T' denotes the MAN_TRADES, and 'MAN_M' denotes the MAN_MART.

| Attack | None | PGD-40 | AA | FWA-40 | TI-DIM | CW$_2$ | DDN |
|---|---|---|---|---|---|---|---|
| AT | **48.41±1.08** | 17.38±1.68 | 11.30±1.68 | 10.38±0.81 | 29.02±0.81 | 0.00±0.00 | 0.31±0.03 |
| MAN | 43.80±1.61 | **18.17±1.67** | **12.46±1.34** | **13.14±1.78** | **29.41±2.36** | **17.31±1.70** | **4.03±0.21** |
| TRADES | **44.61±0.94** | 19.21±1.95 | 12.80±1.43 | 10.78±1.54 | 30.36±0.84 | 0.00±0.00 | 0.09±0.02 |
| MAN_T | 43.66±0.83 | **20.14±1.56** | **12.87±1.21** | **14.94±1.67** | **30.40±0.69** | **1.89±0.56** | **0.87±0.14** |
| MART | **42.21±1.95** | 20.93±1.21 | 15.79±1.41 | 12.99±0.55 | 30.81±0.88 | 0.00±0.00 | 0.04±0.01 |
| MAN_M | 42.20±1.07 | **21.23±1.16** | **15.85±1.29** | **15.12±1.43** | **31.34±1.03** | **1.69±0.37** | **0.63±0.12** |

In addition, we evaluate the adversarial accuracy of defense methods against white-box adaptive attacks on *CIFAR-10* by using the VggNet-19 as the target model and the transition network. As shown in Table 6, our method still achieves better performance.

Table 6: Adversarial accuracy (percentage) of defense methods against white-box adaptive attacks on *CIFAR-10*. The target model is VggNet-19.

| Attack | None | PGD-40 | AA | FWA-40 | TI-DIM | CW$_2$ | DDN |
|--------|------|--------|------|--------|--------|--------|------|
| AT | **80.91** | 29.83 | 26.00 | 7.55 | 49.47 | 0.10 | 0.15 |
| MAN | 80.22 | **37.18** | **33.20** | **17.88** | **49.75** | **38.13** | **6.10** |

## B.2 SCENARIO (II)

In scenario (ii), we design an adversarial attack to destroy the crucial transition matrix of our defense. Since the ground-truth transition matrix is not given, we use the target attack strategy to craft adversarial examples. We choose an anti-diagonal identity matrix as an example of the target transition matrix in the target attack. The target transition matrix $T^*$ is shown in Fig 4. Note that there may be other attacks that can be designed, but this is beyond the scope of our work, and we would not explore further in this paper.

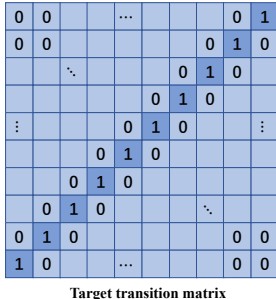

Target transition matrix

Figure 4: Target transition matrix $T^*$ for the target adversarial attack.

## B.3 SCENARIO (III)

In scenario (iii), the optimization goal of the target attack can be designed as:

$$\max_{\tilde{x}} \left[ -\mathcal{L}(\tilde{\boldsymbol{y}} \cdot T(\tilde{x}; \omega), \boldsymbol{y^*}) + \mathcal{L}_{ce}(\tilde{\boldsymbol{y}}, \boldsymbol{y}) \right], \text{ subject to: } \|x - \tilde{x}\| \le \epsilon, \tag{10}$$

where $\boldsymbol{y^*}$ is the target label in the form of a vector set by the attacker.

Note that we report the *last* adversarial accuracy (i.e., the adversarial accuracy of the last epoch models obtained during training) in Table 1, 2, 3, 4 and 5 following the discussion in Carmon et al. (2019), instead of the *best* results.

## B.4 DEFENSE TRANSFERABILITY AGAINST ADAPTIVE ATTACKS

We transfer our defense model to VggNet-19 for defending against adaptive attacks. The original target model is ResNet-18. As shown in Table 7, the results show that our method has some transferability against adaptive attacks. We find that the adversarial accuracy of VggNet-19 is lower than that of ResNet-18. However, this shows the importance and effectiveness of the joint adversarial training on the target model and the transition network in our method.

Table 7: Adversarial accuracy (percentage) of our defense method for different target models on *CIFAR-10* against adaptive adversarial attacks.

| Model | PGD-40 | AA | FWA-40 | TI-DIM | CW$_2$ | DDN |
|-------|--------|------|--------|--------|--------|------|
| ResNet-18 | 44.84 | 39.31 | 29.70 | 58.88 | 46.36 | 10.73 |
| VggNet-19 | 19.67 | 13.73 | 11.60 | 13.51 | 17.32 | 5.19 |

## C  THE INFLUENCE OF MORE MODEL PARAMETERS

We think that the improvement of the adversarial robustness has little relationship with the fact that the transition network introduces more model parameters. The transition network is only used to learn the transition matrix, and it does not directly learn the logit output for the instance to predict the class label. The main reason why our method can improve the robustness is that we explicitly model adversarial noise in the form of the transition matrix. We can use this transition matrix to infer the natural label. In addition, considering that the model parameters of our method are indeed larger than those of AT, we conduct the following experiment to compare AT and our method in the case of similar number of model parameters.

Considering that both the target model and the transition network in our method are mainly based on ResNet-18 on *CIFAR-10*, we use two parallel ResNet-18 as the target model to classify the input. We use the average of their logit outputs as the final logit output. We use the training mechanism of AT to train the new target model. In this way, The number of model parameters of our method is similar to that of AT. As shown in Table 8, the results show that our method still has higher adversarial accuracy. This demonstrates that the improvement of the adversarial robustness is not due to the increase of model parameters.

Table 8: Adversarial accuracy (percentage) of defense methods against white-box adaptive attacks on *CIFAR-10*. The defense models have a similar number of model parameters.

| Attack | None | PGD-40 | AA | FWA-40 | TI-DIM | $CW_2$ | DDN |
|--------|------|--------|------|--------|--------|--------|------|
| AT | 83.57 | 42.10 | 38.70 | 13.22 | 56.66 | 0.00 | 0.03 |
| MAN | 82.71 | **44.84** | **39.31** | **29.70** | **58.88** | **46.36** | **10.73** |

