# OpenReview forum: "Modeling Adversarial Noise for Adversarial Defense"
_ICLR.cc/2022/Conference — ICLR 2022 Submitted_

### Official Review · Reviewer_Ee6e · 2021-10-30

**Correctness:** 3
**Technical Novelty And Significance:** 2
**Empirical Novelty And Significance:** 3
**Recommendation:** 3
**Confidence:** 4

**Main Review:**

Strengths:
1. In the experiments part, the authors clearly evaluate the robustness of the models under three adaptive attack scenarios and the empirical results are technically convincing.

Weaknesses:
1. The idea of using the transition matrix seems to come from the widely used methodology in noisy label learning. The authors should discuss those literatures in the related work and compare the difference between the proposed method and those methods. Otherwise, the novelty of the paper can not be well justified.
2. In Section 3.2, the use of mathematical symbols is a bit confusing, and there are situations where symbols are not defined. For example, the sentence $Y'$  denotes the variable for the natural label, and $Y$ denotes the variable for the natural labels” is confusing. The symbols such as $\widetilde{x_i}$, $\widetilde{y_i}$ are not defined. The authors should well organize this part.
3. For the robustness evaluation, In Table 1, the comparison with only AT is not enough. The transition matrix can be a general module combined with more adversarial training methods such as Trades, MART or ect. to conduct more experiments to benchmark the robustness improvement with different adversarial training methods.
4. For  the experiments in Defense transferability, can the authors provide more results against adaptive attacks? Can the transferability exist under the stronger adaptive attacks?
5. For the effectiveness of the transition matrix, I am concerned with the generalization of the learned transition matrix. From the results in Table 1, we can observe that the improvement of the adversarial accuracy is diverse under different attacks, the improvement under AA is just 0.3 and under TI-DIM is 1.76 which is smaller than 2.46 under PGD40 (the test attack is the same with training attack).




**Summary Of The Paper:**

This paper proposes an adversarial defense method MAN by learning an instance-dependent transition matrix exploiting the transition relationship of adversarial labels and natural labels.  MAN generates the transition matrix by training a parameterized deep neural network which takes the mixture data (adversarial and natural data) as input and outputs the instance-dependent transition matrix. During inference, we can infer the natural labels with the adversarial labels and the transition matrix.  The authors evaluate MAN under three adaptive attacks scenarios and show the improved adversarial accuracy with the proposed method.

**Summary Of The Review:**


Overall, from my point of view, I think the novelty of the methodology and experiments evaluation of the paper in current form is not enough. I suggest rejection.

---

> ### Author Response · Authors · 2021-11-20
> **Response to Reviewer Ee6e's comments (2)**
>
> **Defense transferability against adaptive attacks.**
> We transfer our defense model to other naturally trained target model (e,g, VggNet-19) for defending against adaptive attacks. The original target model is ResNet-18. As shown in Table 2, the results show that our method has some transferability against adaptive attacks. We find that the adversarial accuracy of VggNet-19 is lower than that of ResNet-18. However, this shows the importance and effectiveness of the joint adversarial training on the target model and the transition network in our method.
>
> Table 2. Adversarial accuracy (percentage) of our defense model for different target models on *CIFAR-10* against adaptive adversarial attacks.
>
> | Model     | PGD-40 |   AA  | FWA-40 | TI-DIM | CW$_2$ |  DDN  |
> |-----------|:------:|:-----:|:------:|:------:|:------:|:-----:|
> | ResNet-18 |  44.84 | 39.31 |  29.70 |  58.88 |  46.36 | 10.73 |
> | VggNet-19 |  19.67 | 13.73 |  11.60 |  13.51 |  17.32 |  5.19 |
>
> **Generalization to different attacks.** Since the mechanisms and strengths of adversarial attacks are different, the improvement of the adversarial accuracy against different attacks is typically different. As shown in Table 1, the representative adversarial training methods (e.g., TRADES and MART ) also show different improvements against multiple attacks. We will further improve the generalization of our method against different attacks in future work, especially against strong attacks (e.g., AA, FWA and CW$_2$).

---

> > ### Author Response · Authors · 2021-11-22
> > **Response to Reviewer Ee6e's comments (3)**
> >
> > Thank you again for your efforts to review our paper. We have answered all the questions. If there is anything unclear, we will address it further.

---

> ### Author Response · Authors · 2021-11-20
> **Response to Reviewer Ee6e's comments (1)**
>
> Thank you for your time and efforts spent on reviewing our paper. Your thorough main review and comments are very important to the improvement of our work, which we address below:
>
> **Transition matrix in label-noise learning.** In label-noise learning, the transition matrix is used to infer clean labels from given noisy labels. The transition matrix denotes the probabilities that **clean** labels flip into **noisy** labels [1,2,3]. The label-noise learning methods utilize the transition matrix to correct the training loss on noisy data (i.e., the clean instance with the **noisy label**). Given the noisy class posterior probability, the clean class posterior probability can be obtained, i.e.,$p(\boldsymbol{y}|x)=(T(x)^{\top})^{-1} p(\boldsymbol{y}^{*}|x)$, where $\boldsymbol{y}^{\*}$ denotes the noisy label in the form of a vector. Differently, in our method, the transition matrix is used to infer natural labels from observed adversarial labels. The transition matrix denotes the probabilities that **adversarial** labels flip into **natural** labels. We utilize the transition matrix to help compute the training loss on the adversarial instance with the **natural** label. For the observed adversarial class posterior probability, the natural class posterior probability can be obtained, i.e., $p(\boldsymbol{y}|\tilde{x})=T(\tilde{x})^{\top}p(\tilde{\boldsymbol{y}}|\tilde{x})$. In addition, our method for the first time utilizes the transition matrix to explicitly model adversarial noise for improving adversarial robustness. We will supply the related work about the transition matrix in label-noise learning in the revised version.
>
> [1] Liu, Tongliang, and Dacheng Tao. Classification with noisy labels by importance reweighting. IEEE T-PAMI, 2015.
>
> [2] Xia, Xiaobo, et al. Part-dependent label noise: Towards instance-dependent label noise. ANIPS, 2020.
>
> [3] Yang, Shuo, et al. Estimating Instance-dependent Label-noise Transition Matrix using DNNs. arXiv:2105.13001, 2021.
>
> **Symbol typo.**
> We are sorry for the symbol typos. We would correct the typos: "$Y^{\prime}$ denotes the variable for the mixture labels, and $Y$ denotes the variable for the natural labels". In addition, the symbols $\tilde{x}_i$ and $\tilde{y}_i$ have been defined in *Problem setting*: "Let $\widetilde{X}$ and $\widetilde{Y}$ be the variables for adversarial instances and adversarial labels respectively. We denote by $\\{(\tilde{x}_i, \tilde{y}_i)\\}\_{i=1}^{n}$ the adversarial data drawn according to a distribution of the variables $(\widetilde{X},\widetilde{Y})$."
>
> **Compare with other adversarial training methods.**
> Our proposed method is applicable to different adversarial training methods. To avoid the bias caused by different adversarial training methods, we apply our method to TRADES [4] and MART [5], and compare it with TRADES and MART on *CIFAR-10* and *Tiny-ImageNet*. We use PGD-10 to craft adversarial training data. As shown in Table 1, the results show that our method improves the adversarial accuracy. We find that using TRADES affects the performance against L$_2$-norm based attacks. We will further solve this issue in future work.
>
>
> Table 1. Adversarial accuracy (percentage) of defense methods against white-box adaptive attacks. The target model is ResNet-18.
>
> | Dataset       | Attack     |  None | PGD-40 |   AA  | FWA-40 | TI-DIM | CW$_2$ |  DDN |
> |---------------|------------|:-----:|:------:|:-----:|:------:|:------:|:------:|:----:|
> | CIFAR-10      | TRADES     | **80.68** |  49.38 | 44.43 |  23.92 |  57.45 |  0.00  | 0.16 |
> |               | MAN_TRADES | 76.28 |  **49.94** | **44.49** |  **30.51** |  **57.53** |  **7.69**  | **1.18** |
> |               | MART       | **76.15** |  50.93 | 44.37 |  25.63 |  59.23 |  0.00  | 0.02 |
> |               | MAN_MART   | 75.29 |  **50.97** | **44.45** |  **31.24** |  **59.28** |  **6.08**  | **1.06** |
> | Tiny-ImageNet | TRADES     | **44.61** |  19.21 | 12.80 |  10.78 |  30.36 |  0.00  | 0.09 |
> |               | MAN_TRADES | 43.66 |  **20.14** | **12.87** |  **14.94** |  **30.40** |  **1.89**  | **0.87** |
> |               | MART       | **42.21** |  20.93 | 15.79 |  12.99 |  30.81 |  0.00  | 0.04 |
> |               | MAN_MART   | 42.20 |  **21.23** | **15.85** |  **15.12** |  **31.34** |  **1.69**  | **0.63** |
>
> [4] Zhang, Hongyang, et al. Theoretically principled trade-off between robustness and accuracy. ICML, 2019.
>
> [5] Wang, Yisen, et al. Improving adversarial robustness requires revisiting misclassified examples. ICLR, 2019.

---

> > ### Comment · Reviewer_Ee6e · 2021-11-28
> > **Thank you for the response**
> >
> > Thank you for the response and the additional experimental results.
> >
> > After reading the response, I notice that the improvement is marginal when the method is combined with TRADES and MART (e.g. 0.6% improvement on CIFAR-10 with TRADES against PGD40 and 0.06% against AA). In current status, the experiments are not solid enough to demonstrate the effectiveness of the proposed method. Thus, I prefer to retain my score.

---

> > > ### Author Response · Authors · 2021-11-28
> > > **Response to Reviewer Ee6e**
> > >
> > > Thank you for reviewing our response. Although our method has only a small improvement on PGD-40 and AA, it has a significant improvement on the strong attack that use geometric information (e.g.,  FWA), and attacks based on different norms (e.g., L2-norm CW). In addition, the adversarial accuracy against DDN is increased from 0.09% to 10.73% compared with AT. We believe that these improvements may be valuable to the adversarial learning community.

---

### Official Review · Reviewer_9XAu · 2021-10-31

**Correctness:** 3
**Technical Novelty And Significance:** 3
**Empirical Novelty And Significance:** 2
**Recommendation:** 5
**Confidence:** 5

**Details Of Ethics Concerns:**

I find no ethics concerns.

**Main Review:**

Strengths:
1. The defense strategy that embedding an instance-dependent transition matrix into the target model is novel.
2. The experiments show that the performance of the proposed framework is better than normal adversarial training (AT).

Weakness:
1. The authors claim that the proposed framework can be viewed as adversarial training and the transition matrix can be embedded into the target model. Thus, the authors should conduct experiments by embedding the transition matrix into current adversarial training strategies, e.g., [a,b]. Such experiments are necessary to demonstrate the effectiveness of the proposed method, and should not be viewed as the future work.
[a] Zhang, Hongyang, et al. "Theoretically principled trade-off between robustness and accuracy." International Conference on Machine Learning. PMLR, 2019.
[b] Zheng, Yaowei, Richong Zhang, and Yongyi Mao. "Regularizing neural networks via adversarial model perturbation." Proceedings of the IEEE/CVF Conference on Computer Vision and Pattern Recognition. 2021.

2. Experiments in Table 1 and 2 should also be conducted with other network structures besides ResNet18.

3. The trained model with MAN will lose the performance on the clean samples without perturbations. Especially, the distance is obvious for tiny-ImageNet as shown in Table5. This lead me to doubt the effectiveness of MAN.

4. The size of transition matrix will increase a lot with the increase of class number. Thus, this method would cause greater cost for the dataset with larger class number, e.g., ImageNet.

5. Why the training epoch is different for MAN and the baseline (150 vs 100)?


**Summary Of The Paper:**

This paper proposes to learn a transition relationship between adversarial labels and natural labels for achieving adversarial defense. Especially, the authors introduced an instance-dependent transition matrix, which can be seamlessly embedded with the target model, to relate the adversarial labels and natural labels. Experiments on CIFAR and tiny-ImageNet demonstrate the proposed defense framework can outperform normal adversarial training.

**Summary Of The Review:**

Although the proposed method is novel by embedding a transition matrix to the target model, there are several concerns:
1) the experiments with more kinds of AT baselines,

2) the decrease of performance on clean samples,

3) the applicability of the proposed method for the dataset with large class number,

4) the comparison fairness.

Thus, I think this paper is not enough for publication until these concerns are solved.

---

> ### Author Response · Authors · 2021-11-20
> **Response to Reviewer 9XAu's comments （1）**
>
> Thank you for your time and efforts spent on reviewing our paper. Your thorough comments are very important to the improvement of our work, which we address below:
>
> **Embed the transition matrix into other adversarial training methods.**
> We apply our method to TRADES [1] and MART [2] on *CIFAR-10*. As shown in Table 1, the results show that our method improves the adversarial accuracy. We find that using these adversarial training methods affects the performance against L$_2$-norm based attacks. We will further solve this issue in future work.
>
> [1] Zhang, Hongyang, et al. Theoretically principled trade-off between robustness and accuracy. ICML, 2019.
>
> [2] Wang, Yisen, et al. Improving adversarial robustness requires revisiting misclassified examples. ICLR, 2019.
>
> Table 1. Adversarial accuracy (percentage) of defense methods against white-box adaptive attacks. The target model is ResNet-18.
>
> | Dataset       | Attack     |  None | PGD-40 |   AA  | FWA-40 | TI-DIM | CW$_2$ |  DDN |
> |---------------|------------|:-----:|:------:|:-----:|:------:|:------:|:------:|:----:|
> | CIFAR-10      | TRADES     | **80.68** |  49.38 | 44.43 |  23.92 |  57.45 |  0.00  | 0.16 |
> |               | MAN_TRADES | 76.28 |  **49.94** | **44.49** |  **30.51** |  **57.53** |  **7.69**  | **1.18** |
> |               | MART       | **76.15** |  50.93 | 44.37 |  25.63 |  59.23 |  0.00  | 0.02 |
> |               | MAN_MART   | 75.29 |  **50.97** | **44.45** |  **31.24** |  **59.28** |  **6.08**  | **1.06** |
> | Tiny-ImageNet | TRADES     | **44.61** |  19.21 | 12.80 |  10.78 |  30.36 |  0.00  | 0.09 |
> |               | MAN_TRADES | 43.66 |  **20.14** | **12.87** |  **14.94** |  **30.40** |  **1.89**  | **0.87** |
> |               | MART       | **42.21** |  20.93 | 15.79 |  12.99 |  30.81 |  0.00  | 0.04 |
> |               | MAN_MART   | 42.20 |  **21.23** | **15.85** |  **15.12** |  **31.34** |  **1.69**  | **0.63** |
>
> **Other network structures.**
> We evaluate the adversarial accuracy of defense methods against white-box adaptive attacks on*CIFAR-10* by using the VggNet-19 as the target model and the transition network. As shown in Table 2, our method still achieves better performance.
>
> Table 2. Adversarial accuracy (percentage) of defense methods against white-box adaptive attacks on *CIFAR-10*. The target model is VggNet-19.
>
> | Attack |  None | PGD-40 |   AA  | FWA-40 | TI-DIM | CW$_2$ |  DDN |
> |--------|:-----:|:------:|:-----:|:------:|:------:|:------:|:----:|
> | AT     | **80.91** |  29.83 | 26.00 |  7.55  |  49.47 |  0.10  | 0.15 |
> | MAN    | 80.22 |  **37.18** | **33.20** |  **17.88** |  **49.75** |  **38.13** | **6.10** |
>
> **Performance on the natural instances.**
> The adversarial training used in this paper indeed reduces the accuracy of natural data. As shown in Tabel 1, other representative adversarial training methods also lose the performances on the natural data. Our method significantly improves the accuracy against multiple adversarial attacks (e.g., from 19.44\% to 29.70\% for FWA and from 0.00\% to 46.36\% for CW$_2$), which is expected to be valuable. We will further optimize our method to alleviate the negative effect on natural data.
>
> **More cost for larger class number.**
> For the dataset with larger class number (e.g., *Tiny-ImageNet*), we just increase the number of output nodes of the last fully connected layer in the transition network. Our method would not cause a large increase in cost for the dataset with a large class number.
>
> **Training epoch.**
> Our method introduces a transition network, and conducts joint adversarial training on the target model and the transition network. To make the models fully converge, we increase the training epoch.

---

> > ### Author Response · Authors · 2021-11-22
> > **Response to Reviewer 9XAu's comments （2）**
> >
> > Thank you again for your efforts to review our paper. We have answered all the questions. If there is anything unclear, we will address it further.

---

> > ### Comment · Reviewer_9XAu · 2021-11-28
> > **Comments for authors**
> >
> > Thanks for your responses!
> > After viewing your response, I still have concerns:
> >
> > 1. The proposed methods would largely decrease the performance on the natural instances (e.g., for TRADES and MART on CIFAR10, the decrease is 4.4% and around 1%, respectively. However, the improvement on the adversarial learning is universally small, e.g., for TRADES and MART on CIFAR10, the improvement on PGD-40, AA, TI-DIM, and DDN is not large enough.
> >
> > 2. VGG is not SOTA network structures, and the authors should conduct experiments with more SOTA structures.
> >
> > In consideration of the fact that the improvement is not obvious, I will not change my score.

---

> > > ### Author Response · Authors · 2021-11-28
> > > **Response to Reviewer 9XAu's comments**
> > >
> > > Thank you for reviewing our response. Although our method has only a small improvement on PGD-40, AA and TI-DIM, it has a significant improvement on the strong attack that use geometric information (e.g.,  FWA), and attacks based on different norms (e.g., L2-norm CW). In addition, the adversarial accuracy against DDN is increased from 0.09% to 10.73% compared with AT. We believe that these improvements may be valuable to the adversarial learning community.

---

### Official Review · Reviewer_aEYo · 2021-11-02

**Correctness:** 2
**Technical Novelty And Significance:** 2
**Empirical Novelty And Significance:** 2
**Recommendation:** 3
**Confidence:** 3

**Main Review:**

Strengths
1.	Overall speaking the paper is well written and the method is clearly presented.
2.	The proposed approach seems novel to me. The method is empirically validated by using several attack methods that not only attack the classifier but also the transition network as well.

Weaknesses
1.	I am not fully convinced by the motivation. I would like to see supports on the claim that data pre-processing based methods suffer from high dimensionality problem and addressing the dimensionality problem in modeling adversarial noise is good for robustness. In addition, supposed that modeling perturbations in low-dimensional space is beneficial, it is still not clear that the proposed low-dimensional transition matrix actually models the adversarial noise. It would be helpful if the authors can address these concerns.
2.	It seems like the proposed transition network works as a kind of post-processing. Instead of parameterizing $p(y|x)$ using a neural network $f(x)$, the proposed method parameterizes $p(y|x)$ using $g_{\omega}(x, h_{\theta}(x))$. Following this lens, why cannot we take $g_{\omega}(x, h_{\theta}(x))$ as a whole and expect a single function $f(x)$ to learn and perform as the same? Please kindly elaborate if I am mistaken.
3.	In experiments, the comparison with SOTAs may not be fair. Does the method improve robustness merely because it uses a more complicated neural network architecture with more capacity (as it introduces an additional network $g_{\omega}$ )? It would be great if the authors can align the number of network parameters used by each method in the comparison.

Other questions: How does the proposed method perform combined with TRADES[1]?

[1] Hongyang Zhang, Yaodong Yu, Jiantao Jiao, Eric P. Xing, Laurent El Ghaoui, and Michael I. Jordan. Theoretically principled trade-off between robustness and accuracy. In ICML, 2019.


**Summary Of The Paper:**

This paper proposes a defense method based on Modeling Adversarial Noise. The authors introduce an additional transition network to capture the transition matrix from adversarial predictions to vanilla predictions. The final output is the rectified prediction using this transition matrix. The proposed method is novel to me and is empirically validated by the authors under several adaptive attacks.


**Summary Of The Review:**

Given the points listed, I give the current rating here. It would be helpful if the questions listed can be addressed.

---

> ### Author Response · Authors · 2021-11-20
> **Response to Reviewer aEYo's comments (2)**
>
> **Introduce more model parameters**. We think that the improvement of the adversarial robustness has little relationship with the fact that $g_{\omega}$ introduces more model parameters. The transition network $g_{\omega}$ is only used to learn the transition matrix, and it does not directly learn the logit output for the instance to predict the class label. The main reason why our method can improve the robustness is that we explicitly model adversarial noise in the form of the transition matrix. We can use this transition matrix to infer the natural label. In addition, considering that the model parameters of our method are indeed larger than those of AT, we conduct the following experiment to compare AT and our method in the case of similar number of model parameters.
>
> Considering that both the target model and the transition network in our method are mainly based on ResNet-18 on *CIFAR-10*, we use two parallel ResNet-18 as the target model to classify the input. We use the average of their logit outputs as the final logit output. We use the training mechanism of AT to train the new target model. In this way, The number of model parameters of our method is similar to that of AT. As shown in Table 1, the results show that our method still has higher adversarial accuracy. This demonstrates that the improvement of the adversarial robustness is not due to the increase of model parameters.
>
> Table 1. Adversarial accuracy (percentage) of defense methods against white-box adaptive attacks on *CIFAR-10*. The defense models have a similar number of model parameters.
>
> | Attack |  None | PGD-40 |   AA  | FWA-40 | TI-DIM |  CW$_2$  |  DDN  |
> |--------|:-----:|:------:|:-----:|:------:|:------:|:-----:|:-----:|
> |   AT   | **83.57** |  42.10 | 38.70 |  13.22 |  56.66 |  0.00 |  0.03 |
> |   MAN  | 82.71 |  **44.84** | **39.31** |  **29.70** |  **58.88** | **46.36** | **10.73** |
>
> **Compare with TRADES.**
> Our proposed method is applicable to different adversarial training methods. To avoid the bias caused by different adversarial training methods, we apply our method to TRADES and compare it with TRADES on *CIFAR-10* and *Tiny-ImageNet*. We use PGD-10 to craft adversarial training data. As shown in Table 2, the results show that our method improves the adversarial accuracy. We find that using TRADES affects the performance against L$_2$-norm based attacks. We will further solve this issue in future work.
>
> Table 2. Adversarial accuracy (percentage) of defense methods against white-box adaptive attacks. The target model is ResNet-18.
>
> | Dataset       | Attack     |  None | PGD-40 |   AA  | FWA-40 | TI-DIM |  CW$_2$ |  DDN |
> |---------------|------------|:-----:|:------:|:-----:|:------:|:------:|:----:|:----:|
> | CIFAR-10      | TRADES     | **80.68** |  49.38 | 44.43 |  23.92 |  57.45 | 0.00 | 0.16 |
> |               | MAN_TRADES | 76.28 |  **49.94** | **44.49** |  **30.51** |  **57.53** | **7.69 | **1.18** |
> | Tiny-ImageNet | TRADES     | **44.61** |  19.21 | 12.80 |  10.78 |  30.36 | 0.00 | 0.09 |
> |               | MAN_TRADES | 43.66 |  **20.14** | **12.87** |  **14.94** |  **30.40** | **1.89** | **0.87** |

---

> > ### Author Response · Authors · 2021-11-22
> > **Response to Reviewer aEYo's comments (3)**
> >
> > Thank you again for your efforts to review our paper. We have answered all the questions. If there is anything unclear, we will address it further.

---

> > > ### Comment · Reviewer_aEYo · 2021-11-29
> > > **Comments for the authors**
> > >
> > > Thank the authors for the responses!
> > >
> > > From my perspective, some core arguments made by the authors, e.g., "the proposed transition matrix actually models the adversarial noise", are not very convincing: they are more of something that the authors hope to be true but not something that can be proven or demonstrated. If the robustness improvement is significant enough, I would tend to consider those arguments as possible interpretations of why the proposed method works. However, I found that all the reviewers have concerns regarding the empirical significance and some of these have not been properly addressed yet.
> > >
> > > 1. The authors only conduct experiments on resnet18 and VGG. VGG is not a SOTA architecture for adversarial robustness; Wideresnet is more prevalent.
> > >
> > > 2. The proposed method decreases the clean accuracy significantly so it is hard to judge whether the comparison is fair.
> > >
> > > The above two concerns are raised by reviewer 9XAu and have not been completely addressed in the rebuttal yet. I would suggest the authors add experiments on Widesnet and draw the trade-off curves between clean ACC and robust ACC of both MAN+TRADES and TRADES by tuning the weight of the KL term for comparison.
> > >
> > > 3. In the additional experiments using similar model parameters, it is intriguing that under PGD-40 attacks, AT with parallel ResNet-18 performs even worse than AT using a single ResNet-18 (42.10% v.s. 42.38% as reported by the authors). An ensembled model or a model with more parameters is often expected to perform better. May you further explain it?
> > >
> > > Given the listed, I tend to retain my score.

---

> ### Author Response · Authors · 2021-11-20
> **Response to Reviewer aEYo's comments (1)**
>
> Thanks for your time and efforts spent on reviewing our paper. Your thorough main review and comments are very important to the improvement of our work, which we address below:
>
> **Supports on high dimensionality problem.** Pre-processing based defenses typically denoise the adversarial instance or squeeze its feature. Denoising based defenses need to generate a high-dimensional instance (e.g., 32$\times$32$\times$3 dimensions on *CIFAR-10*) for each adversarial instance. The recovered instance may contain residual adversarial noise [1], and have problems that generative models usually suffer (e.g., texture loss and deformation). Feature squeezing based methods may remove useful information and cause the squeezed instance to be obviously inconsistent with the real natural instance [2], thus affecting the performance of the target model. We provide some similar illustrations and references in the *Introduction* and *Preliminaries*.
>
> [1] Liao, Fangzhou, et al. Defense against adversarial attacks using high-level representation guided denoiser. In CVPR, 2018.
>
> [2] Xu, Weilin, David Evans, and Yanjun Qi. Feature squeezing: Detecting adversarial examples in deep neural networks. arXiv:1704.01155, 2017.
>
> **Supports on addressing the dimensionality problem.** In this paper, our aim is to avoid the high-dimensional problems. Our method is to learn an instance-dependent label transition matrix. The transition matrix can describe how label flips for a given adversarial data and thus can model the adversarial noise information. The dimensionality of the transition matrix is typically lower than the processed instance and is easier to learn. In addition, the adversarial noise controls the flip from natural labels to adversarial labels. Modeling adversarial noise in the label space can help directly correct the adversarial label, which is beneficial to improve the adversarial accuracy of the target model. We thus propose to model adversarial noise in the low-dimensional label space.
>
> **Not clear that the proposed transition matrix actually models the adversarial noise.** The transition matrix describes how label flips for a given adversarial data and thus models the adversarial noise information. Specifically, adversarial noise has feature which controls the flip from the natural label to the adversarial label. The transition matrix encode the probabilities that adversarial labels flip into natural labels. It can describe the transition relationship opposite to the flip controlled by adversarial noise. Thus, the transition matrix can be used to explicitly model the feature of adversarial noise.
>
> **Why not take $g_{\omega}(\tilde{x}, h_{\theta}(\tilde{x}))$ as a whole and expect a single function to learn?**
> We will elaborate the mechanism of our method more clearly and concisely below to address the confusion:
>
> Generally, we can parameterize $p(y|x)$ for the natural data $(x,y)$ by using a neural network (i.e., the target model $h_{\theta}(x)$ in our paper). However, for the adversarial instance $\tilde{x}$, the predicted label $\tilde{y}$ (adversarial label) is typically wrong, i.e., we obtain $p(\tilde{y}|\tilde{x})$ for the adversarial instance $\tilde{x}$ by using $h_{\theta}(\tilde{x})$. To correct the adversarial label $\tilde{y}$ back to the natural label $y$, we design the transition network $g_{\omega}(\tilde{x})$ to generate the transition matrix $T(\tilde{x};\omega)=g_{\omega}(\tilde{x})$. We can obtain $p(y|\tilde{x})$ by exploiting $T(\tilde{x};\omega)$: $p(y|\tilde{x})=T(\tilde{x};\omega)^{\top}p(\tilde{y}|\tilde{x})$.
>
> Note that the input for the transition network $g_{\omega}(\cdot)$ is only the instance $\tilde{x}$ (or $x$), excluding $h_{\theta}(\tilde{x})$. The target model $h_{\theta}(\cdot)$ and the transition network $g_{\omega}(\cdot)$ are in parallel. We parameterize $p(y|\tilde{x})$ by using $g_{\omega}(\tilde{x})^{\top} \cdot \delta(h_{\theta}(\tilde{x}))$ instead of $g_{\omega}(\tilde{x}, h_{\theta}(\tilde{x}))$, where $\delta$ is the softmax function. Of course, we can take the transition network $g_{\omega}(\cdot)$ and the target model $ h_{\theta}(\cdot)$ as a whole and use a function $f(\cdot)$ to obtain $p(y|\tilde{x})$: $p(y|\tilde{x})$=$f_{\theta,\omega}(\tilde{x})$, where $f_{\theta,\omega}(\tilde{x})=g_{\omega}(\tilde{x})^{\top} \cdot \delta(h_{\theta}(\tilde{x}))$. The training method for learning $f(\cdot)$ is the same as our method. In fact, in the section *defending against adaptive attacks*, we treat the target model and the transition network as a whole to conduct adversarial training. In addition, the adversarial training (e.g.,  the AT in our paper) directly learns $p(y|\tilde{x})$ via a simple function $h_{\theta}(\cdot)$ parameterized by the target model. However, the experiments show that the method has lower adversarial accuracy than our method.

---

### Decision · Program_Chairs · 2022-01-20

**Decision:**

Reject

**Comment:**

This paper aims to model adversarial noise by learning the transition relationship between adversarial labels and natural labels. In particular, an instance-dependent transition matrix to relate adversarial labels and natural labels. Reviewers agreed that the paper is well motivated and well written, and the proposed method is novel. Meanwhile, reviewers raised some concerns about experiments and paper presentation. During discussion, the authors provided a lot of additional results that partially addressed the reviewers' concerns. However, the reviewers still think the experimental part of this paper should be further strengthened before acceptance.

Thus, I recommend to reject this paper. I encourage the authors to take the review feedback into account and submit a future version to another venue.